# Observation of crystallisation dynamics by crystal-structure-sensitive room-temperature phosphorescence from Au(I) complexes

Yuki Kuroda[1], Masakazu Tamaru[1], Hitoya Nakasato[1], Kyosuke Nakamura[1], Manami Nakata[1], Kyohei Hisano [1], Kaori Fujisawa[1] & Osamu Tsutsumi [1✉]

The aggregation behaviour of Au(I) complexes in condensed phases can affect their emission properties. Herein, aggregation-induced room-temperature phosphorescence (RTP) is observed from the crystals of trinuclear Au(I) complexes. The RTP is highly sensitive to the crystal structure, with a slight difference in the alkyl side chains causing not only a change in the crystal structure but also a shift in the RTP maximum. Furthermore, in nanocrystals, reversible RTP colour changes are induced by phase transitions between crystal polymorphs during crystal growth from solution or the pulverisation of bulk crystals. The colour change mechanism is discussed in terms of intermolecular interactions in the crystal structure of the luminescent aggregates. The results suggest that the behaviour in nanocrystals may differ from that in bulk crystals. These insights will advance the fundamental understanding of crystallisation mechanisms and may aid in the discovery of new materials properties for solids with nano- to micrometre sizes.

[1] Department of Applied Chemistry, Ritsumeikan University, 1-1-1 Nojihigashi, Kusatsu 525-8577, Japan. ✉email: tsutsumi@sk.ritsumei.ac.jp

Organic nanocrystals, which enable enhanced performance of practical devices and manifest novel functionalities not observed in their bulk states, have attracted considerable attention in various modern industries[1–3]. With recent developments in the fabrication procedures for organic nanocrystals, the crystal size can now be easily controlled from tens of nanometres to bulk samples. A fascinating feature of nanocrystals is the dependence of the materials properties on the crystal size, with drastic differences observed between single molecules and bulk states. Thus, to design nanocrystals with specific functionalities, it is crucial to control the crystallisation process and the resulting crystal size. However, the crystallisation processes of organic compounds, especially in the early stages of crystallisation and the polymorphic behaviour during crystal growth, remain unclear[4–6].

Numerous theoretical and experimental studies have attempted to reveal the dynamics of crystallisation processes using methods such as electron microscopy, IR spectroscopy, and Raman spectroscopy; however, it is difficult to observe the crystallisation processes directly in situ and in real time during the early stage, which hinders the elucidation of the crystallisation and polymorphic behaviour during crystal growth[7–12]. A more powerful method is the in situ observation of luminescence changes during the crystallisation process[12]. However, organic luminescent materials show efficient luminescence in very dilute solutions, but

the aggregation of molecules in concentrated solutions and in solid states results in decreased luminescence, which is known as aggregation-caused quenching (ACQ)[13,14]. ACQ is a general phenomenon for most organic luminescent molecules. Over the past two decades, materials in which the aggregation of molecules enhances the radiative decay of excited states have been developed, and this phenomenon is recognised as aggregation-induced emission (AIE)[15–24]. Many types of organic AIE-active molecules (AIEgens) have been found to be non-emissive in dilute solutions but emissive in condensed phases[20–24]. We consider that AIEgens may allow crystallisation processes to be visualised because of the drastic enhancement in luminescence intensity caused by the formation of aggregates.

Some Au(I) complexes exhibit AIE activity owing to an intermolecular interaction known as the aurophilic interaction[25,26], in which a non-covalent Au–Au bond is formed between molecules. Because of this interaction, the luminescence of Au(I) complexes can be increased by aggregation in condensed phases[27–46]. We hypothesised that the introduction of multiple aurophilic interactions could further enhance the AIE activity of Au complexes[36–46] and enable us to visualise a crystallisation process clearly. In addition, as the luminescence arises from the aggregates, the luminescence properties of Au complexes should be quite sensitive to the aggregate structure[27–46]. Consequently, the luminescence properties can be altered by changing the aggregate structure[29–35].

In this study, we synthesise three Au complexes bearing alkyl side chains with slightly different lengths. Using these complexes, we determine the relationship between the crystal structure and the luminescence properties in terms of the intermolecular interactions and then utilise this relationship to investigate the crystallisation process from solution. We find that highly efficient room-temperature phosphorescence (RTP) is emitted from the trinuclear Au complex crystals and that the colour of the luminescence is extremely sensitive to the crystal structure. Furthermore, the crystal structures of some of the investigated Au complexes depend on the crystal size; thus, the luminescence colour can be altered by varying the crystal size, allowing us to observe the early stage of the crystallisation process and the polymorphic behaviour during crystal growth.

## Results

**Luminescence of bulk crystals**. Three trinuclear Au(I) complexes (DT4–DT6) bearing n-alkyl side chains were synthesised (Fig. 1a, Supplementary Methods, Supplementary Figs. 1 and 2). The crystals of the complexes were placed between a pair of quartz plates to observe their photoluminescence. When the crystals were irradiated with UV light (254 nm) at room temperature (RT, 15 °C) in the presence of air, bright visible photoluminescence could be observed by the naked eye (Fig. 1b). The crystals of each complex exhibited a different luminescence colour (yellow for DT4, purple for DT5, and red for DT6). However, none of the complexes showed photoluminescence in dilute solutions ($<10^{-4}$ mol L$^{-1}$), which indicates that the complexes are AIE active and that the photoluminescence is emitted from aggregates (Supplementary Fig. 6). The AIE activity was further confirmed in a mixture of good and poor solvents (CH$_2$Cl$_2$/methanol) at different volume ratios (Supplementary Note 1 and Supplementary Fig. 8).

The photoluminescence and excitation spectra of the complex crystals are shown in Fig. 1c and Supplementary Figs. 6 and 7, and the corresponding photophysical parameters are summarised in Table 1. Two luminescent bands appeared in the spectrum of DT4 and DT5 crystals; thus, the photophysical parameters were determined separately for each band. The luminescence lifetime

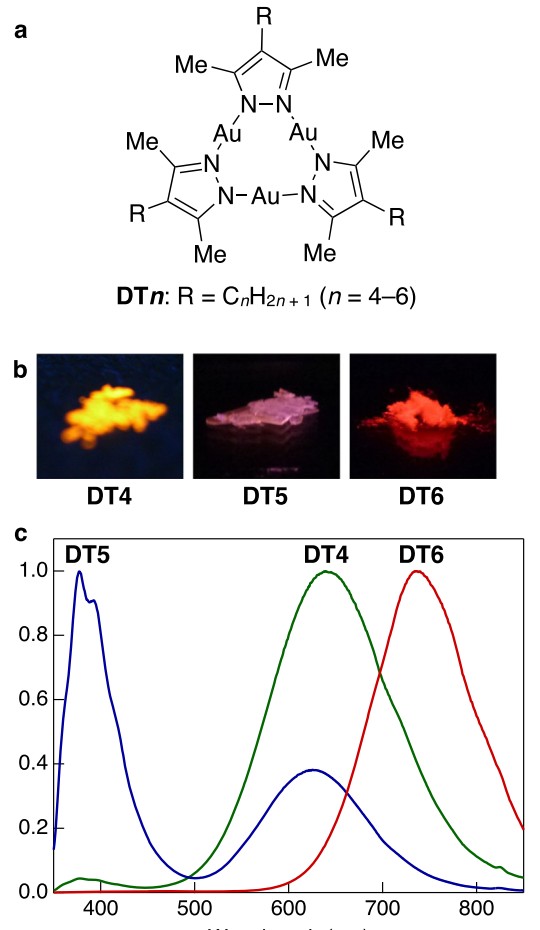

**Fig. 1 Structures and photoluminescence properties of the Au(I) complexes used in this study. a** Molecular structures of trinuclear Au(I) complexes DT4-DT6. **b** Photographs of the photoluminescence emitted by crystals of DT4-DT6 under UV irradiation (254 nm) at RT in the presence of air. **c** Photoluminescence spectra (excitation at 280 nm) of DT4-DT6 crystals at RT.

**Table 1 Photophysical parameters for photoluminescence in crystals at RT by excitation at 280 nm[a].**

| | Bulk crystals[b] | | | Nanocrystals[e] | | |
|---|---|---|---|---|---|---|
| | $\lambda_{max}^{lum}$ (nm) | $\Phi$ (%)[c] | $\tau$ (µs)[d] | $\lambda_{max}^{lum}$ (nm) | $\Phi$ (%)[f] | $\tau$ (µs)[g] |
| DT4 | 378 (minor) | 2[h] | 0.09 (0.96)7.6 (0.04) | 736 | 31 | 11 |
| | 638 (major) | 56[h] | 0.06 (−0.76)9.8 (1.0) | | | |
| DT5 | 377 (major) | 10[h] | 3.5 (0.65)6.5 (0.35) | 736 | 44 | 14 |
| | 630 (minor) | 3[h] | 4.5 (−0.28)23 (1.1) | | | |
| DT6 | 738 | 75 | 12 | 727 | 69 | 15 |

[a]$\lambda_{max}^{lum}$ maximum RTP wavelength, $\Phi$ RTP quantum yield, $\tau$ RTP lifetime.
[b]Crystals with a size of >100 µm obtained by recrystallisation from $CH_2Cl_2$/acetone.
[c]Estimated by integration in the wavelength range from 350 to 450 nm for the DT4 minor band, from 450 to 875 nm for the DT4 major band, from 350 to 500 nm for the DT5 major band, from 500 to 875 nm for the DT5 minor band, and from 350 to 875 nm for DT6.
[d]The $\tau$ in bulk crystals was measured at 390 nm for DT4 minor band, 615 nm for DT4 major band, 405 nm for DT5 major band, 605 nm for DT5 minor band, and 700 nm for DT6, respectively. The pre-exponential factors are indicated in the parentheses, and the negative factor means rise of the luminescence.
[e]Crystals with a size of ~100 nm obtained by reprecipitation from THF/water, and the measurements were performed on a filter.
[f]Estimated by integration in the wavelength range from 500 to 850 nm.
[g]Measured at 700 nm.
[h]Normalised RTP quantum yield[47].

($\tau$) of complexes were measured at each luminescence band (Supplementary Note 1 and Supplementary Figs. 9–11). At the major luminescence band, the complexes had $\tau$ on the microsecond timescale at RT, indicating clearly that RTP was observed. Furthermore, the crystals of each complex showed a relatively high RTP quantum yield ($\Phi$), even in the presence of air (total $\Phi = 13$–75%) owing to both the heavy atom effect of the Au atoms and the crystallisation enhancement effect on phosphorescence[15–19,48,49]. Interestingly, the maximum RTP wavelength ($\lambda_{max}^{lum}$) was highly sensitive to the length of the alkyl side chain, as shown in Fig. 1b, c. The luminescent centres in DT4–DT6 consist of Au atoms and the π-electron systems of pyrazole rings. Typically, the length of the alkyl side chains would not be expected to affect the electronic structure of such luminescent centres. To understand the origin of the different RTP colours, a single-crystal X-ray structure analysis was performed (Supplementary Data 1–3); the key crystallographic data are summarised in Supplementary Table 1 and selected geometrical parameters are summarised in Supplementary Table 2. Each complex crystallised in a different space group, and supramolecular polymers were formed in the crystals (Supplementary Fig. 3). As shown in Fig. 2, a pair of closest neighbour molecules that form a dimer was extracted from each crystal structure to indicate the intermolecular interactions clearly. The selected intermolecular distances are summarised in Supplementary Table 2. The interatomic distances between the Au atoms in neighbouring molecules were ~3.6 Å or less in the DT4 and DT6 crystals, which suggests the existence of a weak interaction between Au atoms (aurophilic interaction)[25,26]. Moreover, the distances between the Au atoms and the pyrazolyl ring centroid were in the range of 3.6–3.8 Å, and the angle between the vector normal to the pyrazolyl ring and that passing through the centroid to the Au atom ($\theta$) was >20° in DT5 and DT6. These geometrical parameters also suggest the presence of an intermolecular Au–π interaction in the DT5 and DT6 crystals[50–53]. The observed intermolecular interactions are depicted in Fig. 2, namely, a Au–Au interaction at a single site in DT4, a Au–π interaction at two sites in DT5, and both interactions, each at two sites, in DT6. We consider these dual modes and multisite intermolecular interactions in the crystals to play a crucial role in determining the RTP colour of the aggregates.

To clarify the mechanistic origin of the different RTP colours in the crystals, the molecular orbitals, transition energies, and oscillator strengths ($f$) of the complexes were calculated using time-dependent density functional theory (TD-DFT;

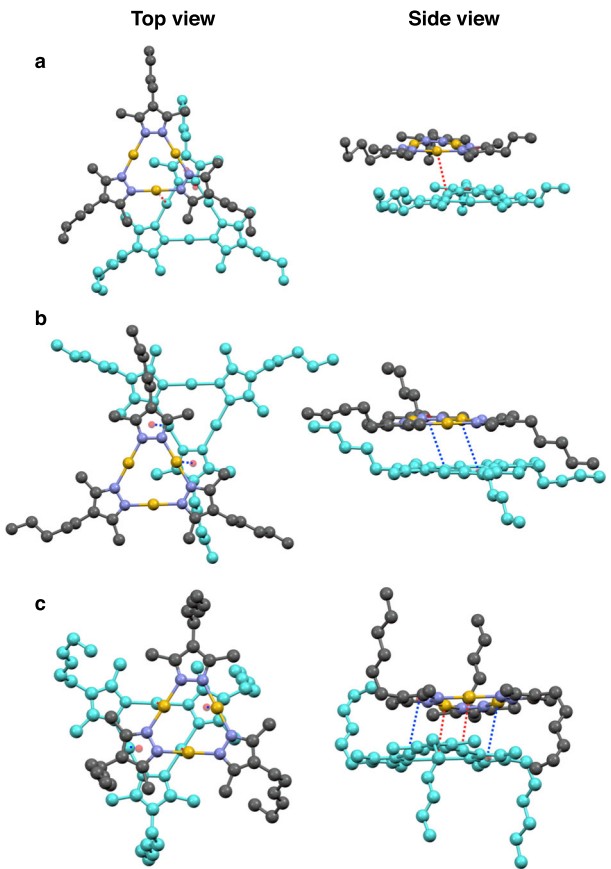

**Fig. 2 Crystal structures of the Au(I) complexes.** Crystal structures of **a** DT4, **b** DT5, and **c** DT6. For each complex, the structure of a pair of closest neighbour molecules was extracted. For clarity, hydrogen atoms are omitted. Only the molecules in the foreground are shown in colour (grey, C; purple, N; yellow, Au; pink, pyrazole ring centroid), whereas the molecules in the background are shown in light blue. Intermolecular interactions are indicated with broken lines (red, Au–Au interaction; blue, Au–π interaction).

Supplementary Note 1, Supplementary Figs. 13–15 and Supplementary Table 3). The calculations were performed for dimers formed in the crystals using the arrangements shown in Fig. 2. The TD-DFT calculations suggested that electronic transitions from

the ground state ($S_0$) to a singlet excited state ($S_n$) are located in the UV region between 254 and 262 nm. For comparison, the calculation was performed also for the monomer of a model complex (Supplementary Fig. 16 and Supplementary Table 5). In the monomer, the transitions are located at 244 nm, and this calculated result is roughly consistent with the absorption spectra of the complexes in dilute solutions (Supplementary Fig. 6). Thus, it can be concluded that 10–20-nm shift of the absorption wavelength was induced by dimer formation.

The calculation results for the DT6 dimer, which have previously been reported, suggested that the electronic transitions in the excitation spectrum can be assigned to a transition from the ligand-based π-orbital to intermolecular non-covalent aurophilic bonding, i.e., a ligand-to-metal–metal charge transfer (LMMCT) transition[36–46]. Taking into account the allowed transitions with relatively large $f$ values ($f > 10^{-2}$), the LMMCT transition was also the main electronic transition in the DT4 dimer. In contrast, in the DT5 dimer, the calculations suggested a transition from the π-orbital to intramolecular aurophilic bonding because DT5 showed no intermolecular Au–Au interaction in the ground state. Similar luminescence spectra have been reported for trinuclear Au complexes, with the crystals exhibiting an emission band with a vibronic structure at ~350 nm and two bands without vibronic structures at ~650 and ~750 nm, depending on the measurement temperature[36–43]. Yang and co-workers proposed that the structured band at ~350 nm is due to the monomeric triplet emission of complexes with a strong ligand character, and that the broad unstructured bands at ~650 and ~750 nm originate from metal-centred excimeric triplet states with a different effective symmetry[40]. Since the results obtained in this study are consistent with these previous reports, we can conclude that the emission at each band is from the same excited state reported previously.

The excitation energy from the $S_0$ to the lowest triplet excited state ($T_1$) was also calculated for the same structure of dimers, and the results suggested that the $S_0$–$T_1$ transitions are located at around 300 nm (Supplementary Table 4). As shown in Supplementary Fig. 6, a large excitation band appeared in the crystals at this wavelength. Omary et al. proposed that this excitation band observed in Au complex crystals represents spin-forbidden transitions[41]. The experimental and computational results obtained in this study are consistent with this previous report. Thus, we conclude that the excitation band observed at ~300 nm can be assigned to the direct $S_0$–$T_1$ transition. The large Stokes shift in the RTP, especially observed in DT4 and DT6 crystals, indicate significant excited-state distortions for the RTP state of the complexes[12].

From above discussion based on the luminescence spectra, structural analysis, and computational results, we can conclude that a slight difference in the alkyl side-chain length causes a change in the crystal structure and the intermolecular interactions, resulting in a modified emission mode and inducing a large spectral change, as shown in Fig. 1.

**Luminescence of nano- and microcrystals**. During observations of the RTP behaviour in the $CH_2Cl_2$/methanol mixed solvent system to confirm the AIE activity, we found that the RTP colour and spectrum of the complexes in the mixed solvent, in which small crystals were suspended, were considerably different from those of the bulk crystals prepared by conventional recrystallisation (Supplementary Note 1 and Supplementary Fig. 8). These results suggest that the RTP colour is also influenced by the crystal size. Therefore, we characterised the RTP behaviour of small crystals. Nanometre-sized crystals were simply obtained by a reprecipitation method, in which the addition of a drop of THF

solution (5.0 mmol L$^{-1}$, 50 μL) to vigorously stirred water (10 mL) produced a nanocrystal suspension. The mean diameters ($\bar{D}$) of the nanocrystals, as determined by dynamic light scattering, were ~100 nm (92 ± 19, 111 ± 28, and 99 ± 18 nm for DT4, DT5, and DT6, respectively; Supplementary Fig. 4). The obtained nanocrystals were stable in water for at least several weeks, and crystal growth, co-aggregation, and coalescence were not observed in the water suspension. The nanocrystals were collected by filtration, and their luminescence behaviour was observed on the filter (Fig. 3a). The nanocrystals showed a single exponential decay, and almost the same lifetimes as the DT6 bulk crystal were obtained (Table 1 and Supplementary Fig. 12); thus, the complexes emitted the RTP in nanocrystals. Interestingly, the nanocrystals of all the complexes exhibited red emission with a similar $\lambda_{max}^{lum}$ (Table 1), whereas the RTP colours of the bulk crystals depended on the length of the alkyl side chains. The DT4 and DT5 nanocrystals showed new luminescence band in the longer wavelength region (Table 1). However, for DT6, the RTP band of the nanocrystals almost matched that of the bulk crystal.

As mentioned above, the dimer structure resulting from the Au–Au and Au–π interactions determined the RTP colour of the DT$n$ crystals. However, the RTP results for the nanocrystals suggest that similar luminescent dimers with the same intermolecular interactions are formed by all the complexes in nanocrystals. Thus, the crystal structures of complexes DT4 and DT5 should be different in the nanocrystals and in the bulk crystals. To confirm this assumption, we performed a powder X-ray diffraction (XRD) analysis (Fig. 3b). For DT6, the same XRD pattern was observed for both the nanocrystals and the bulk crystal. In contrast, the XRD results indicate that the crystal structures of the DT4 and DT5 complexes changed in the nanocrystals. Therefore, we can conclude that in nanometre spaces, these complexes form dimers with the same intermolecular interactions as DT6 and emit red RTP. For further confirmation of this phenomenon, we investigated how the RTP behaviour was affected by gradually changing the crystal size. We controlled the crystal size by changing the combination of solvents used for reprecipitation. For example, using a $CHCl_3$ (5.0 mmol L$^{-1}$, 50 μL)/methanol (10 mL) mixture, micrometre-size crystals were obtained immediately after reprecipitation ($\bar{D}$ = 1.5 ± 0.4, 1.8 ± 0.5, and 1.8 ± 0.6 μm for DT4, DT5, and DT6, respectively; Supplementary Fig. 5). Unlike the size stability observed for the nanocrystals in water, the microcrystals grew in the methanol suspension through Ostwald ripening, and after 24 h, crystals with sizes of 50–100 μm and the same morphology as the corresponding bulk crystals were obtained. All the complexes also showed red RTP just after reprecipitation, i.e. ~1 μm crystal size. However, during the crystal growth process, the RTP spectra and the colours of DT4 and DT5 changed to those of the corresponding bulk crystals. After 24 h, the RTP spectra were the same as those of the bulk crystals, as shown in Fig. 3c for DT4 as a representative example (see Supplementary Fig. 17 for the other complexes, and Supplementary Fig. 18 for detailed time evolution of the RTP in DT4). From these experiments, we can conclude that complexes DT4 and DT5 showed crystalline polymorphism, and that a phase transition between polymorphs occurred during crystal growth. The powder XRD analysis supports that the crystal structures of DT4 and DT5 were altered during crystal growth (Supplementary Note 1 and Supplementary Fig. 20). In contrast, the RTP spectrum of DT6 was unchanged, even after the crystals grew to a size of 50 μm.

Although we believe that the dependence of the RTP colour change on the crystal size was caused by the phase transition between crystalline polymorphs during crystal growth, solvent effects on both nucleation and crystal growth should be considered (Supplementary Note 1 and Supplementary Fig. 19)

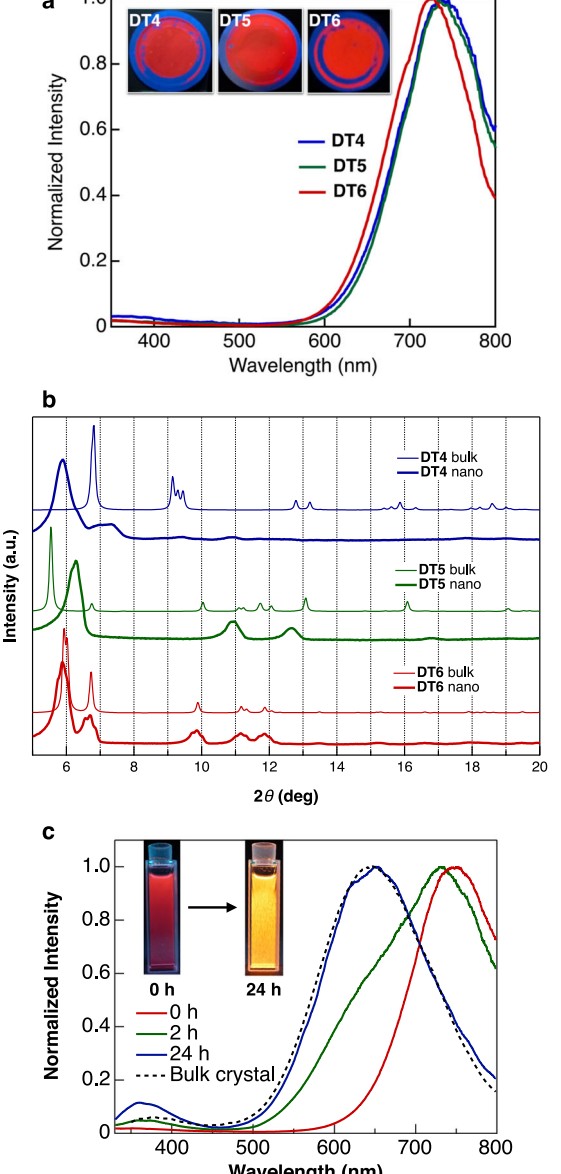

**Fig. 3 Photoluminescence and structural properties of the nanocrystals. a** Photoluminescence spectra (excitation at 280 nm) of **DT*n*** nanocrystals. The insets show photographs of the nanocrystals on filters taken under irradiation at 254 nm. **b** Experimental powder XRD patterns of nanocrystals (bold line) and XRD patterns of bulk crystals simulated from the single-crystal X-ray structures (thin line). **c** Time evolution of the photoluminescence spectra (excitation at 280 nm) of DT4 crystals in a methanol suspension prepared by precipitation from CHCl₃/methanol (50 μL/10 mL). The insets show photographs of DT4 crystal suspensions taken under irradiation at 254 nm.

[54,55]. Thus, we prepared micro- and nanocrystals by mechanical pulverisation of the bulk crystals with a planetary ball mill. The pulverised crystals were filtered twice using filters with different pore sizes to separate the crystals by size; this procedure is shown schematically in Fig. 4a. Figure 4b shows size histograms for the pulverised crystals of DT5 as a representative example. The as-pulverised crystals had a bimodal size distribution ($\bar{D} = 1.5 \pm 1.4$ μm and $80 \pm 23$ nm). After filtration of the methanol suspension of the pulverised crystals using an 8 μm pore-size filter (Filtrate 1), the size distribution was still bimodal ($\bar{D} = 1.0 \pm 0.60$ μm and $67 \pm 17$ nm). However, the proportion of large crystals decreased.

Further filtration of Filtrate 1 using a 0.8 μm pore-size filter (Filtrate 2) gave a monomodal size distribution ($\bar{D} = 132 \pm 85$ nm). Figure 4c shows a comparison of the RTP spectra of the methanol suspensions of the as-pulverised DT5 crystals and the crystals in the filtrates normalised at 740 nm. All the crystals showed two main RTP bands corresponding to the nanocrystals at 740 nm and the bulk crystal at 380 nm. The relative intensity of the band at 380 nm decreased following repeated filtration, indicating that the crystals with a smaller size obtained by pulverisation showed the same properties as the nanocrystals obtained by reprecipitation (Fig. 3a). In Fig. 4d, e, the RTP spectra of the crystals in the filtrate and the residue on the filter are compared for each filtration process. After the first filtration process, the RTP spectrum of the residual crystals on the filter (Residual 1, $\bar{D} > 8$ μm) was similar to that of the bulk crystal; however, the RTP spectrum of Filtrate 1 exhibited a relatively strong band corresponding to the nanocrystals. Following the second filtration process, the residual crystals (Residual 2, $\bar{D} > 0.8$ μm) still showed a spectrum similar to that of the bulk crystal. In contrast, for Filtrate 2, the RTP band corresponding to the nanocrystals was predominant, whereas only a very small RTP band corresponding to the bulk crystal was observed. Thus, although the nanocrystals were prepared from the bulk crystal, the crystal structure of the complex changed, and the RTP behaviour showed a dependence on the crystal size. The threshold size for this phenomenon was estimated as ~10 μm.

## Discussion

In the bulk crystals, the investigated complexes showed different RTP colours, caused by different crystal structures, depending on the length of the alkyl side chains. In contrast, in the nanocrystals, the same RTP colour was observed for all the complexes. Therefore, we propose that polymorphs exist for the DT4 and DT5 crystals. According to the DFT calculations, among the investigated complexes, DT6 exhibits the most stable dimer because of the multiple Au–Au and Au–π intermolecular interactions (the association energy for dimer formation was 20.8 kJ mol⁻¹ in DT4, 19.1 kJ mol⁻¹ in DT5, and 34.5 kJ mol⁻¹ in DT6; Supplementary Table 2). In general, in bulk crystals, the minimisation of the free energy of the entire system does not necessarily require that the most stable dimer structure is adopted, and materials occasionally show conformational polymorphism[56]. During the crystallisation process, there is a competition between a positive surface free-energy change and a negative bulk free-energy change, which creates a barrier for the growing crystals, as shown schematically in Fig. 4f. The radius corresponding to the maximum free energy is defined as the critical nucleus radius; when the crystal (cluster) radius exceeds this value, crystal growth occurs. According to the Ostwald step rule, in the initial stage, a metastable polymorph of the bulk crystal is formed kinetically because the nucleation barrier is lower than that of the stable polymorph of the bulk crystal[56]. Until the crystal radius reaches the crossing point of the energy curves of the polymorphs, the metastable phase is the thermodynamically stable phase. Then, at the crossing point, a metastable-to-stable crystal phase transition takes place. Based on this argument, the phenomena observed in this study can be interpreted as follows. The nanocrystals of DT4 and DT5 were the metastable phase, in which the intermolecular interactions as well as the dimer structures were the same as those in the DT6 bulk crystal. Subsequent crystal growth in the suspension induced the metastable-to-stable crystal phase transition at the crossing point of the energy curves. Considering the general balance between the surface free energy and the bulk free energy, the crossing point for thermodynamic stability is in the nanoscale region;[57,58] however, for the investigated complexes, the phase transition took place at a crystal radius >~10 μm. The association

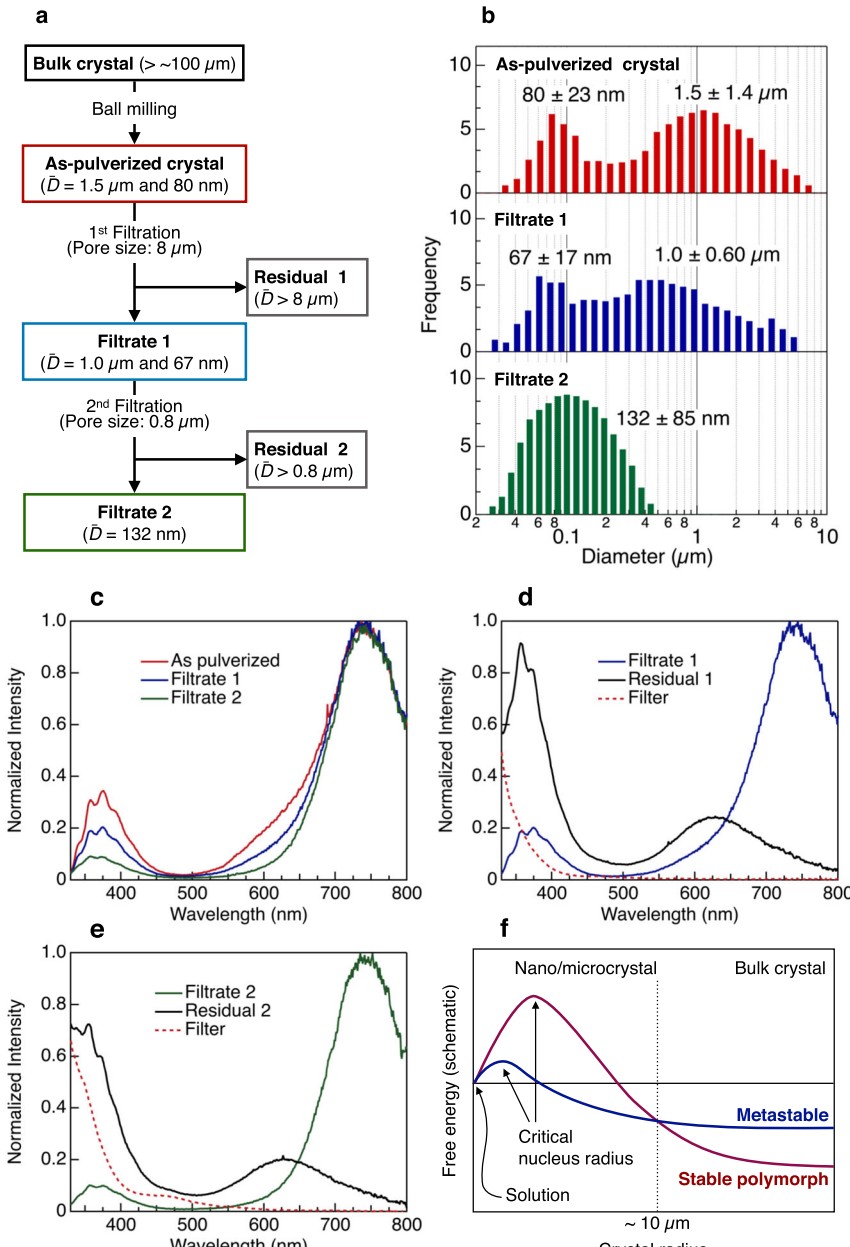

**Fig. 4 Preparation and properties of pulverised micro-/nanocrystals. a** Flow chart for micro-/nanocrystal preparation by mechanical pulverisation. **b** Size distributions of pulverised crystals observed by dynamic light scattering. **c** Luminescence spectra (excitation at 250 nm) of pulverised crystals before and after filtration. **d, e** Luminescence spectra (excitation at 250 nm) of pulverised crystals after filtration: **d** first filtration and **e** second filtration. The spectra of residual crystals on the filter (black lines) and the background fluorescence from the filter (red dashed lines) are also shown. **f** Schematic representation of the free energies of two different polymorphs as a function of crystal size in DT4 and DT5: blue, metastable polymorph observed in nano- and microcrystals; red, stable polymorph observed in the bulk crystal.

energies of the DT4 and DT5 dimers in the nanocrystals should be similar to that of the DT6 dimer; therefore, this relatively large association energy, i.e., large negative bulk free energy, of the nanocrystals stabilises the metastable phase. Thus, the crossing point is shifted to a larger radius than in the general crystallisation system. This relatively large negative bulk energy may also lower the energy barrier for the nucleation of the metastable crystal. As a result, the nucleation of the metastable crystal occurred much faster than that of the stable crystal, and RTP was only observed from the metastable crystal at small crystal radii, namely, during the early stage of crystallisation. When the bulk crystal was pulverised, the reverse phase transition between polymorphs occurred. In some crystals, mechanical stress influences the crystal structure[59]. During the

pulverisation process, the structures of all the crystals in a ball-milling pot might be affected by the mechanical force. Generally, it is very difficult to prepare crystals with sizes of <10 μm by hand grinding using a mortar and pestle[60]. Considering that luminescent mechanochromism can be induced by hand grinding of common mechanochromic materials, the ball-milling process used in this study should supply sufficient mechanical energy for the obtained microcrystals to show a RTP colour change. Nevertheless, the ball-milled crystals with a size of >8 μm (Residual 1) showed the same RTP behaviour as the original bulk crystals. These results indicate that the mechanical force effects can be disregarded in the DT*n* crystals. Thus, we conclude that the observed colour

changes as well as the crystal-structure changes were caused by the crystal sizes.

In conclusion, we found that the Au(I) complexes DT*n* showed efficient RTP in crystals, even in the presence of air. However, the RTP was very sensitive to the crystal structures. Small change in the alkyl side-chain length, which caused an alteration of the crystal packing structure, had a great effect on the RTP colour of the bulk crystals, even though the primary structure of the luminescent centre remaining unchanged. Furthermore, we showed that the crystal structure and RTP colour of DT4 and DT5 changed depending on the crystal size. These phenomena indicate that some materials show completely different behaviour in small spaces (<10 μm in the case of DT4 and DT5). In many devices, nanometre- or micrometre-sized materials are used, for example, as particles or thin films. Thus, the results of this study suggest that in such cases, the behaviour of the materials may not be the same as that of the bulk crystals, and attention must be paid to the size of materials. Although crystal formation is an essential process in materials applications as well as in fundamental science, this process is still not understood completely. Based on the phenomena observed in this study, crystallisation processes could be observed directly in situ on the molecular level using luminescence colour changes. Thus, the dynamic processes involved in crystallisation, i.e., nucleation, crystal size evolution over time, and associated phase transitions between polymorphs, could be observed directly in real time and in situ using luminescence spectroscopy. We consider these phenomena to be useful for advancing the fundamental understanding of crystallisation mechanisms in solution and also expect that new materials properties will be revealed for nano- to micro-sized solids.

## Methods

**Material preparation**. The Au complexes were synthesised according to the literature[37,46]. Each product was purified on a silica gel column and then recrystallised from a $CH_2Cl_2$/acetone mixture to give bulk crystal of the complex (Supplementary Methods).

**Photophysical properties**. The UV–visible absorption and steady-state photoluminescence spectra were recorded on a JASCO V-550 absorption spectrophotometer and a Hitachi F-7500 fluorescence spectrophotometer, respectively. The quantum yields of photoluminescence were determined using a Quantaurus-QY absolute photoluminescence quantum yield spectrometer (C11347-01, Hamamatsu). The photoluminescence decay profiles were measured using a Quantaurus-Tau photoluminescence lifetime measurement system (C1136-21, Hamamatsu) at an excitation wavelength of 280 nm.

**Preparation of nano-/microcrystals**. A THF solution of DT*n* (5.0 mmol $L^{-1}$, 50 μL) was added to 10 mL of deionised water under vigorous magnetic stirring at 1400 rpm at RT to obtain DT*n* nanocrystals. The resultant nanocrystals were collected by filtration of the suspension with a filter (pore size, 0.1 μm; 025010MFPES, Asone). Using the same method with a $CHCl_3$/methanol solvent system, crystals with a diameter of ~1 μm were obtained just after reprecipitation. Nanocrystals were also prepared by ball milling using a planetary ball mill (PULVERISETTE 7, Fritsch). DT*n* bulk crystals (20 mg), a methanol/water mixture (2/5, v/v; 1.2 mL), and zirconia balls (diameter, 1.0 mm; 303 mg) were placed in a 1.5 mL glass vial. The vial was capped and set in a stainless steel pot, and the crystals were ball-milled at 320 rpm for 48 h. After milling, the crystals were separated by size using an 8 μm pore-size filter (K800A025A, Advantec) and then a 0.8 μm pore-size filter (K080A025A, Advantec).

**Crystal size determination**. The size and size distribution of the nanocrystals were determined by dynamic light scattering analysis using a DLS-8000 scatterometer (Otsuka). The sizes of crystals >10 μm were directly observed with a polarised optical microscope (BX51, Olympus).

## Data availability

All data generated or analysed during this study are included in this article (and its Supplementary Information files. The X-ray crystallographic coordinates for structures reported in this Article have been deposited at the Cambridge Crystallographic Data

Centre (CCDC), under deposition number CCDC 1973710 for DT4, CCDC 1973709 for DT5, and CCDC 1910566 for DT6[46]. These data can be obtained free of charge from The Cambridge Crystallographic Data Centre via www.ccdc.cam.ac.uk/data_request/cif.

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

## Acknowledgements

We acknowledge technical support from Prof. Kei Ameyama and Dr. Mie Ota (Ritsumeikan Univ.) for ball milling, and from Prof. Fuyuki Ito (Shinshu Univ.) for nanocrystal characterisation. This research was supported by JSPS KAKENHI (18K05265 and 18H03764 for O.T.; 19K21131 for K.H.), JST A-STEP (JPMJTM19C9), the JICA CKP/ARC (2018-01), the Japan–Egypt Research Cooperative Program (JSPS/MOSR-STDF), the Ritsumeikan Global Innovation Research Organization (R-GIRO), and the Cooperative Research Program of the Network Joint Research Centre for Materials and Devices (Tokyo Institute of Technology).

## Author contributions

O.T. conceived the project. Y.K., M.T., H.N., K.N., and M.N. performed the synthesis of complexes, sample preparation, and the characterisation of nano-/microcrystals. K.F. measured and analysed the single-crystal structures. Y.K., M.T., H.N., K.N., M.N., K.F., and K.H measured the photophysical behaviour of the complexes. K.F., K.H., and O.T. analysed the data and performed the theoretical calculations. Y.K., K.F., K.H., and O.T. prepared the paper, incorporating the contributions of all authors.

## Competing interests

The authors declare no competing interests.
