## [Peer Review File · Communications Chemistry]

Reviewers' comments:

Reviewer #1 (Remarks to the Author):

The authors report about three Au(I) complexes. After characterization, these bulk crystals were observed to have aggregation-induced room-temperature phosphorescence. With the slight differences in their chemical structure, it was found that their luminescence color changed significantly. From experiments, they found the color changes were induced by phase transition between crystal polymorphs. The designed experiments are comprehensive and precise in order to support their explanations. This project is a good example to illustrate in situ observation of luminescence changes during the crystallization process. Therefore, I would like to recommend its publication in Chemical Communications after the following minor corrections.

1. According to the luminescence color changes, crystal structures and theoretical calculations are used to explained this in terms of the intermolecular interactions. To gain more insights about the photophysical pictures of this, it is suggested to do some photophysical measurement in different viscosity and possibly at low temperature, e.g. 77K.
2. From Figure 3c, it illustrates the shift in maximum emission wavelength according to time progression. It is suggested to give more details in between 2 and 24 hours. If the data has been collected already, is there any reason to hide it from the graph?
3. For DT6, it is found to have similar luminescence spectra both in nano and bulk scale. Since the given photoluminescence spectra are normalized, how is the changes in terms of intensity for DT6 from nano to bulk scale?
4. Despite changes in emission wavelength, to gain a better understanding of the phase transition, it is suggested to give decay profiles for room-temperature phosphorescence in nano crystals also.

Reviewer #2 (Remarks to the Author):

In this article the authors report some crystal size dependent photoluminescence of few gold complexes. The results are interesting, but the article needs to address following points:

1. In the title the authors refers the emission as phosphorescence but throughout the manuscript the authors refer it with the general term photoluminescence. Other than the lifetime is there any evidence to assign the emission to phosphorescence.
2. Clear dual emissions are observed in DT5 and DT4. Therefore, at least two emitting states exist. The authors need to address this important observation. I think the shorter wavelength emission is from the monomer and the longer wavelength is from the dimer.
3. To calculate the shift in the band maxima, it is better to compare the shorter wavelength band with shorter wavelength band and the longer wavelength band with longer wavelength band.
4. Since the emissions are from different states the quantum yield should be calculated separately and decays should be measured for both the emissions.
5. Authors may perform TDDFT calculations to obtain the emission spectra of the monomers and the dimers to support the experimental observations.
6. Why the excitation spectra are different from the absorption spectra?
7. How the alkyl chain length affects the spectral characteristics?
8. Why the spectral characteristics of DT4 are in between those of DT5 and DT6?

Reviewer #3 (Remarks to the Author):

In this manuscript, Tsutsumi and co-authors study the photoluminescence behavior of trigold pyrazolate clusters in the view of crystal size and the crystal packing. Three complexes were considered, one has been reported earlier by the same group (Molecules 2019, 24, 4606). The photophysical properties of luminophores in the solid state are often not easy to rationalize, and here the authors have tackled some non-trivial phenomena. First of all, it has been demonstrated that the luminescence of these compounds in bulk crystals can be dramatically changed by means of electronically innocent alkyl side chains. Secondly and more challenging, different emissive behavior of nano/micro/bulk crystals has been analyzed and reasonably correlated with the evolution of crystallization progress and the transformation of polymorphs. In my opinion, this is a valuable contribution, which might of interest to researchers dealing with crystal engineering and photofunctional materials. It can be recommended for publication in Communications Chemistry upon a minor revision.

Comments:

1. Table 1. Two emission wavelengths and two lifetimes should be provided for DT5. Does the quantum yield correspond to the total emission or the HE band only? The excitation spectrum monitored at the LE band (ca. 630 nm) should be also provided (Fig. S6).
2. Page 7 lines 130-132. Generally, the crystallographic distance does not necessarily mean the bonding interaction. Bond energies or bond critical points should be better considered.
3. Page 9. The calculations were performed for the singlet-singlet excitations, but these provide little information about the triplet excited state, from which the emission occurs. I'd recommend adding the geometries of the T1 state.
4. Page 10 last sentence. The statement that the red shift for DT5 is 359 nm is somewhat confusing because the bulk crystals show dual emission, then why the LE band is neglected?
5. ESI Page S13. I'd revise the statement "the luminescence color is highly dependent on the crystal size". It is not just the size but primarily the morphology. DT6 does not show a substantial variation of luminescence with the size of the particle because the packing is not changing.

Reviewer #1

Thank you very much for your valuable comments. We considered your comments very carefully and revised our manuscript on the light of your comments. Our answers to your comments are described in the following:

1. *According to the luminescence colour changes, crystal structures and theoretical calculations are used to explained this in terms of the intermolecular interactions. To gain more insights about the photophysical pictures of this, it is suggested to do some photophysical measurement in different viscosity and possibly at low temperature, e.g. 77K.*

We attempted to observe the photophysical behaviours of the complexes in a solvent with different viscosity; however, it was not possible to do those measurements due to the solubility problems. As described in p. 10–11 (line 182–184) of the manuscript, it has already been reported the luminescence behaviour of similar trinuclear Au complexes at low temperature (ref. 36–41). Similar luminescence spectra have been reported for the trinuclear Au complexes, with the crystals exhibiting an emission band with a vibronic structure at ~350 nm and two bands without vibronic structures at ~650 and ~750 nm, depending on the measurement temperature (ref. 36–43). Yang and co-workers proposed that the structured band at ~350 nm is due to the monomeric triplet emission of complexes with a strong ligand character, and that the broad unstructured bands at ~650 and ~750 nm originate from metal-centred excimeric triplet states with a different effective symmetry (ref. 40). The same luminescence behaviour was also observed for **DT6** at low temperature by Cored *et al.* (ref. 38: *Inorg. Chem.* **57**, 12632–12640 (2018)). Those results support that the luminescent colour of present complexes is determined by the intermolecular interaction as well as the crystal structure; thus, the crystal-structure change, induced by the alkyl-spacer length or by the crystal growth, can alter the luminescence colour.

2. *From Figure 3c, it illustrates the shift in maximum emission wavelength according to time progression. It is suggested to give more details in between 2 and 24 hours. If the data has been collected already, is there any reason to hide it from the graph?*

Thank you for your suggestion. In accordance with your suggestion, we measured the detailed time evolution of the photoluminescence spectra of the microcrystal for **DT4** as a representative example (Fig. S18). This measurement was performed at 18 °C. The crystal growth rate is sensitive to the measurement temperature. At this temperature, the suspension reached the equilibrium almost at 10 h, and the same luminescence spectrum with the bulk

crystal was obtained. In addition, in first 6 h, no significant change in the luminescence spectrum was observed; however, the sharp change in the spectrum was observed between 6 and 10 h. In the suspension, the microcrystals grew gradually through Ostwald ripening. When the crystal radius reaches the crossing point of the energy curves of the polymorphs shown in Fig. 4f, a metastable-to-stable crystal phase transition takes place. Thus, it is reasonable that the sharp spectral change was observed at the specific time between 6 and 10 h, and this sharp change support our conclusion that the luminescence colour change in the present complexes occurred by the crystal–crystal phase transition induced by the crystal growth. This discussion was added to the revised Supplementary Information (p. S15).

3. *For DT6, it is found to have similar luminescence spectra both in nano and bulk scale. Since the given photoluminescence spectra are normalized, how is the changes in terms of intensity for DT6 from nano to bulk scale?*

In Table 1, we compared the quantum yields of RTP between bulk crystal and nanocrystal. In the **DT6** crystals, no significant difference in the quantum yield was observed. This means that the same luminescence intensity was observed in both bulk and nano crystals, if the measurements were performed under the same conditions.

4. *Despite changes in emission wavelength, to gain a better understanding of the phase transition, it is suggested to give decay profiles for room-temperature phosphorescence in nano crystals also.*

The decay profiles of RTP in nanocrystals were added to the Supplementary Information (Fig. S12), and the estimated lifetimes were added to Table 1 in the revised manuscript. To compare the photophysical properties under the same condition, the lifetimes in the all bulk crystals were also remeasured with the same instrument (line 391–392).

The lifetime of the nanocrystals was described in the revised manuscript (line 220–222, p. 12). The RTP lifetimes in the nanocrystals were 11–15 μs , and those are almost the same as it in the **DT6** bulk crystal. Thus, the result supports that the same luminescent aggregates as in the **DT6** bulk crystal were formed in the nanocrystals.

Reviewer #2

Thank you very much for your valuable comments. We considered your comments very carefully and revised our manuscript on the light of your comments. Our answers to your comments are described in the following:

1. *In the title the authors refer the emission as phosphorescence but throughout the manuscript the authors refer it with the general term photoluminescence. Other than the lifetime is there any evidence to assign the emission to phosphorescence.*

The μs lifetimes clearly indicate that the observed luminescence at room temperature is phosphorescence. Thus, we changed the term of “photoluminescence” to room-temperature phosphorescence (RTP).

2. *Clear dual emissions are observed in **DT5** and **DT4**. Therefore, at least two emitting states exist. The authors need to address this important observation. I think the shorter wavelength emission is from the monomer and the longer wavelength is from the dimer.*

Thank you very much for your beneficial suggestions. We have addressed dual emission band in **DT4** and **DT5** in the revised manuscript (line 122–127, p. 7). As described in the manuscript (line 182–184, p. 10–11), Yang and co-workers proposed that the structured band at ~ 350 nm (namely, shorter wavelength emission) is due to the monomeric triplet emission of complexes with a strong ligand character, and that the broad unstructured bands at ~ 650 and ~ 750 nm (namely, longer wavelength emission) originate from metal-centred excimeric triplet states with a different effective symmetry induced by the change in the crystal structure at low temperature⁴⁰. The luminescence spectra, structural analysis, and computational results obtained in this study are consistent with these previous reports. Thus, as you suggested, we concluded that the shorter wavelength emission is from the monomer and the longer wavelength is from the dimer (excimer) in the revised manuscript (line 174–190, p. 10–11).

3. *To calculate the shift in the band maxima, it is better to compare the shorter wavelength band with shorter wavelength band and the longer wavelength band with longer wavelength band.*

Thank you very much for this comment. Thanks to your comment, we realised that the term “shift” is confusing for the readers and unsuitable to express the phenomenon, because actually it is not shift of the luminescence band—the original bands disappeared, and

another new band appeared at the longer wavelength. Therefore, we deleted the term “shift” (line 225–226 in the revised manuscript) and revised the sentence.

4. *Since the emissions are from different states the quantum yield should be calculated separately and decays should be measured for both the emissions.*

We measured the RTP lifetimes separately at each luminescent band (Table 1, Fig. S9–S10) and the results were added to Table 1 in the revised manuscript. To compare the photophysical properties under the same condition, the lifetimes in the **DT6** bulk crystals were also remeasured with the same instrument (line 391–392). In addition, the RTP quantum yields were estimated at each band separately (Table 1).

By the lifetime measurement for **DT4** and **DT5** bulk crystals, we found that both high-energy emission (shorter-wavelength emission) and low-energy emission (longer-wavelength emission) showed bi-exponential decay profiles, and that the rise and decays profile were observed in the low-energy emission. Plausible explanations for these observations were added in the revised Supporting Information (p. S9).

5. *Authors may perform TD-DFT calculations to obtain the emission spectra of the monomers and the dimers to support the experimental observations.*

Thank you very much for your suggestion. We performed the TD-DFT calculation for the monomer and dimers of the Au complex.

In case of monomer, to reduce the calculation load, complex **DT1** was employed as a model compound for the calculation (Fig S16). We employed the standard B3LYP hybrid functionals with SDD (for the Au atoms) and 6-311+G(d,p) (for all other atoms) basis sets for geometry optimization. The vertical excitation energies and oscillator strengths were estimated for the 10 lowest transitions for the optimized equilibrium geometries using TD-DFT with the same hybrid functional and basis set. The calculated transitions with relatively large f values are summarized in Table S5. The DFT calculations suggested that electronic transitions from the ground state to the excited state are located in the UV region at around 244 nm, and these calculated results were roughly consistent with the absorption spectra of the complexes in dilute solution (Fig. S6). In the monomer, the calculations suggested a transition from the π -orbital to intramolecular aurophilic bonding.

In case of dimers, the calculations were performed for dimers formed in the crystals using the arrangements shown in Fig. 2, and the same hybrid functionals and basis sets as the monomer were employed. The TD-DFT calculations suggested that electronic transitions from the ground state (S_0) to the singlet excited state (S_n) are located in the UV region between 254 and 262 nm (Table S3). As mentioned above, the absorption wavelength for

the monomer is calculated at 244 nm. Thus, it can be concluded that 10–20 nm shift of the absorption wavelength was induced by dimer formation.

The excitation energy from the S_0 to T_1 was also calculated for the same structure of dimers (Table S4). The results suggested that the S_0 – T_1 transitions are located at around 300 nm. As shown in Fig. S6, large excitation bands appeared in the crystals at this wavelength. Omary et al. observed the same photophysical behaviour in a similar trinuclear Au complex (ref #41, *Inorg. Chem.* **44**, 8200–8210 (2005)); namely, a weak absorption band with the small molar extinction coefficient (10 – 50 L mol⁻¹ cm⁻¹) appeared at ~ 300 nm in a solution, and the corresponding excitation band was observed also in the crystal at the same wavelength. They proposed that these absorption and excitation bands represent spin-forbidden S_0 – T_n transition. Our experimental and computational results are consistent with their report; thus, we conclude that the excitation bands observed in all complex crystals at ~ 300 nm can be assigned to the direct S_0 – T_n transition. As Omary et al. discussed in ref #41, the large Stokes shift in the RTP, especially observed in **DT4** and **DT6** crystals, indicate significant excited-state distortions for the RTP state of the complexes.

The above discussion was added to the revised manuscript (line 168–173 and 191–200, p. 10–11) and revised Supplementary Information (p. S11).

6. *Why the excitation spectra are different from the absorption spectra?*

The excitation spectra shown in Fig. S6 were measured in the bulk crystal; however, the absorption spectra in the same figure were for the dilute solution. That is the reason why both spectra are not consistent. As stated in our answer for your question #5, Omary et al. proposed that the excitation band at ~ 300 nm is due to the spin-forbidden transition, namely the direct S_0 – T_n transition. Our experimental and computational results also support this proposition; hence, we can consider that this excitation band at ~ 300 nm is due to the spin-forbidden transition. However, since the absorption band in the solution is due to the S_0 – S_n transition—as also stated in our answer for your question #5, the excitation band is different from the absorption band.

7. *How the alkyl chain length affects the spectral characteristics?*

In the dilute solution, there is no effect of the alkyl-chain length on the spectral character (Fig. S6). This means that the alkyl-chain length did not affect the spectral character in the single-molecular level. On the other hand, as shown in Fig. 1, in the bulk crystals, a large change in the luminescence spectra was induced by the alkyl side-chain length. A slight difference in the alkyl-chain length causes a change in both the crystal structure and the intermolecular interactions (Fig. 2), resulting in a modified emission mode.

8. Why the spectral characteristics of DT4 are in between those of DT5 and DT6

As stated above, the spectral-characteristics change was induced by change in both the crystal structure and intermolecular interactions. Depending on the intermolecular interactions, the complexes showed different colour of luminescence. Thus, there is no relationship between the alkyl side-chain length and luminescence wavelength. **DT4** accidentally showed the luminescence wavelength between those of **DT5** and **DT6**.

Reviewer #3

Thank you very much for your valuable comments. We considered your comments very carefully and revised our manuscript on the light of your comments. Our answers to your comments are described in the following:

1. *Table 1. Two emission wavelengths and two lifetimes should be provided for **DT5**. Does the quantum yield correspond to the total emission or the HE band only? The excitation spectrum monitored at the LE band (ca. 630 nm) should be also provided (Fig. S6).*

We measured the RTP lifetimes of **DT4** and **DT5** bulk crystals separately at each luminescent band and the results were added to Table 1 in the revised manuscript. To compare the photophysical properties under the same condition, the lifetimes in the **DT6** bulk crystals were also remeasured with the same instrument (line 391–392). In addition, the RTP quantum yields were estimated at each band (Table 1).

By the lifetime measurement for **DT4** and **DT5** bulk crystals, we found that both high-energy emission and low-energy emission showed bi-exponential decay profiles, and that the rise and decays profile were observed in the low-energy emission. Plausible explanations for these observations were added in the revised Supporting Information (p. S9).

The excitation spectrum monitored at the LE band (620 nm) was also provided in Fig. S7 for **DT5**. When the excitation spectrum was monitored at LE band, no significant change in the excitation wavelength and spectral shape was observed.

2. *Page 7 lines 130-132. Generally, the crystallographic distance does not necessarily mean the bonding interaction. Bond energies or bond critical points should be better considered.*

We estimated the association energy for dimer formation in the complexes by the DFT calculation: 20.8 kJ mol⁻¹ in **DT4**, 19.1 kJ mol⁻¹ in **DT5**, and 34.5 kJ mol⁻¹ in **DT6** (line 318–321 in the revised manuscript, and Table S2). Those association energies reflect the total bond energies of the intermolecular interaction. Generally, the Au–Au interaction is a weak force, comparable to hydrogen bonding, and definitely stronger than standard van der Waals forces. In addition, the Au– π interaction should be weaker than the Au–Au interaction. The tendency of magnitude of the association energies matched the expected magnitude of the intermolecular interactions. Thus, these association energies estimated by the DFT calculations support the formation of the molecular dimer.

3. *Page 9. The calculations were performed for the singlet-singlet excitations, but these provide little information about the triplet excited state, from which the emission occurs. I'd recommend adding the geometries of the T1 state.*

Thank you very much for your suggestion. Additional TD-DFT calculations were performed for the singlet–triplet excitations, and the results were summarized in Table S4. The results suggested that the S_0 – T_1 transitions are located at around 300 nm. As shown in Fig. S6, large excitation bands were observed in the crystals at this wavelength. Omary *et al.* observed the same photophysical behaviour in a similar trinuclear Au complex (ref #41, *Inorg. Chem.* **44**, 8200–8210 (2005)); namely, a weak absorption band with the small molar extinction coefficient (10–50 L mol⁻¹ cm⁻¹) appeared at ~300 nm in a solution, and the corresponding excitation band was observed also in the crystal at the same wavelength. They proposed that these absorption and excitation bands represent spin-forbidden S_0 – T_n transition. Our experimental and computational results are consistent with their report; thus, we conclude that the excitation bands observed in all complex crystals at ~ 300 nm can be assigned to the direct S_0 – T_n transition. As Omary *et al.* discussed in ref #41, the large Stokes shift in the RTP, especially observed in **DT4** and **DT6** crystals, indicate significant excited-state distortions for the RTP state of the complexes.

The above discussion was added to the revised manuscript (line 191–200, p. 11).

4. *Page 10 last sentence. The statement that the red shift for DT5 is 359 nm is somewhat confusing because the bulk crystals show dual emission, then why the LE band is neglected?*

Thank you very much for this comment, and we agree with your comment. Thanks to your comment, we realised that the term “shift” is confusing for the readers and unsuitable to express the phenomenon, because actually it is not shift of the luminescence band—the original bands disappeared, and another new band appeared at the longer wavelength. Therefore, we revised this statement and deleted the term “red shift” (line 225–226 in the revised manuscript). We also mentioned the dual emission of the bulk crystals (line 122–123, p.7) in the revised manuscript.

5. *ESI Page S13. I'd revise the statement “the luminescence colour is highly dependent on the crystal size”. It is not just the size but primarily the morphology. DT6 does not show a substantial variation of luminescence with the size of the particle because the packing is not changing.*

As you suggested, we revised this statement in the revised ESI (line 5–1 from the bottom, p. S15).

REVIEWERS' COMMENTS:

Reviewer #1 (Remarks to the Author):

The authors have adequately revised their manuscript according to my previous comments and suggestions. The quality of the manuscript has been improved after the revision. I do not have further criticism of the work. The revised manuscript is recommended for publication as is.

Reviewer #2 (Remarks to the Author):

The revised manuscript address several queries raised. However, the following information may provide better clarity.

1. Absorption, Emission and excitation spectra as a function of concentration may clarify the formation of dimer.
2. Comparison of excitation spectra at both the band maxima will be useful to understand better.
3. Why the Stokes shift (excitation and emission spectra) of the dimer is very high?
4. Since the absorption spectra are mixture of species the quantum yields reported are normalized emission yields (Photochemistry and Photobiology, 2015, 91: 298–305).

Reviewer #3 (Remarks to the Author):

I find the revision is quite satisfactory and the manuscript can be accepted in its current form.

Reviewer #2

Thank you very much again for your valuable comments. We considered your comments very carefully and revised our manuscript on the light of your comments. Our answers to your comments are described in the following:

1. Absorption, Emission and excitation spectra as a function of concentration may clarify the formation of dimer.

Those spectra for **DT4**, as a representative example, have been added to the ESI (Supplementary Figure 6d). Above the concentration of 10^{-4} mol L⁻¹, a significant shoulder was appeared in the absorption spectra, meaning that the aggregate was formed in the ground state in the solution. Additionally, significant luminescence was observed in the concentration of $> 10^{-4}$ mol L⁻¹, meaning that the luminescence was emitted from the molecular aggregates.

2. Comparison of excitation spectra at both the band maxima will be useful to understand better.

We compared the excitation spectra monitored at both luminescence bands for **DT5** bulk crystal; no significant change in the excitation wavelength and spectral shape was observed. The results are shown in Supplementary Figure 7.

3. Why the Stokes shift (excitation and emission spectra) of the dimer is very high?

We have previously reported that **DT6** crystal showed extremely large Stokes shift in the room-temperature crystals (reference #46), and discussed reasons for the large Stokes shift of the **DT6** bulk crystal in the paper. The complexes show phosphorescence in the crystals. In addition, the luminescence from the crystal is excimeric emission. Those two factors resulted in the large Stokes shift observed in the **DT6** crystals.

4. Since the absorption spectra are mixture of species the quantum yields reported are normalized emission yields (Photochemistry and Photobiology, 2015, 91: 298–305).

The luminescence quantum yields reporting in Table 1 were measured in the crystals of complexes as absolute quantum yields using integration sphere. The suggested paper has described the normalized fluorescence yields measured in the solutions containing several absorbing species, using quinine sulfate in 1 mol L⁻¹ sulfuric acid as a reference. As mentioned above, even when the excitation spectra were monitored at the different emission bands (Supplementary Figure 7), no significant change in the excitation wavelength and spectral shape was observed; this means that only one absorption species existed in the reporting complex crystals. Therefore, in the present crystals, we can determine the absolute emission quantum yields with the integration sphere.

Reviewers' comments:

Reviewer #2 (Remarks to the Author):

If single species is present in the system, the normalized excitation spectra are superimposable, if not they are not superimposable more than one species are present. Supplementary Figure 7 excitation spectra clearly show that both the emissions are from different species. Accordingly, the result and discussion has to be modified. The quantum yield is an intrinsic property of the molecule and is independent of the way it is measured whether it is using reference or integrating sphere. The emission intensity is proportional to absorbance. Since the absorbance is a mixture the reported quantum yields are normalized fluorescence yield as reported in Photochemistry and Photobiology, 2015, 91: 298–305. Measurement of phosphorescence quantum yield required quantum yield of ISC. I recommend to revise the manuscript and highlight the revision in the revised manuscript.

Reviewer #2

Thank you very much again for your valuable comment. We have reconsidered your comment and literatures very carefully, and have also redone minute check of our data. We agree with your comment about the normalized quantum yield. Therefore, we have revised our manuscript on the light of your comment; we clearly indicated that the obtained quantum yields are normalized quantum yields in the footnote of Table 1 and added the literature to references.

Reviewer #2 (Remarks to the Author):

The authors revised the manuscript by taking into account all my comments and now the manuscript is suitable for publication.